# Bending Stability of Ferroelectric Gated Graphene Field Effect Transistor for Flexible Electronics

**DOI:** 10.3390/ma16103798

**Published:** 2023-05-17

**Authors:** Guangliang Hu, Yinchang Shen, Lvkang Shen, Chunrui Ma, Ming Liu

**Affiliations:** 1School of Microelectronics, Xi’an Jiaotong University, Xi’an 710049, China; huguangliang@126.com (G.H.); yinchangshen@stu.xjtu.edu.cn (Y.S.); shenlvkang@mail.xjtu.edu.cn (L.S.); 2State Key Laboratory for Mechanical Behavior of Materials, Xi’an Jiaotong University, Xi’an 710049, China; chunrui.ma@mail.xjtu.edu.cn

**Keywords:** ferroelectric film, GFET, polarization, flexible devices

## Abstract

In this work, we explored the potential of the ferroelectric gate of (Pb_0.92_La_0.08_)(Zr_0.30_Ti_0.70_)O_3_ (PLZT(8/30/70)) for flexible graphene field effect transistor (GFET) devices. Based on the deep understanding of the *V*_Dirac_ of PLZT(8/30/70) gate GFET, which determines the application of the flexible GFET devices, the polarization mechanisms of PLZT(8/30/70) under bending deformation were analyzed. It was found that both flexoelectric polarization and piezoelectric polarization exist under bending deformation, and their polarization direction is opposite under the same bending deformation. Thus, a relatively stable of *V*_Dirac_ is obtained due to the combination of these two effects. In contrast to the relatively good linear movement of *V*_Dirac_ under bending deformation of relaxor ferroelectric (Pb_0.92_La_0.08_)(Zr_0.52_Ti_0.48_)O_3_ (PLZT(8/52/48)) gated GFET, these stable properties of the PLZT(8/30/70) gate GFETs make them have great potential for applications in flexible devices.

## 1. Introduction

In recent years, with the development of flexible microelectronic devices, ferroelectric materials have attracted more and more attention due to their unique polarization reversal characteristics. Compared with conventional organic polymers, ferroelectric materials exhibit excellent thermal and chemical stability [1,2]. In practical applications, flexible electronic devices need to achieve higher speed and lower power consumption, and ferroelectric materials have been considered the most promising research system to achieve these goals. Meanwhile, the ferroelectric gate graphene field effect transistor (GFET) has developed rapidly and has shown great application potential in recent years [3,4,5,6,7]. In particular, the use of ferroelectric materials with large dielectric constants as gate dielectrics in ferroelectric gate-controlled GFET devices not only enhances the regulation of graphene carrier concentration but also enables non-volatile regulation of graphene performance [8,9]. With the rapid development of flexible electronics, the conventional silicon-based ferroelectric gate GFETs are less attractive due to their poor flexibility. Therefore, in order to solve the flexibility problem, researchers have tried to prepare ferroelectric gate GFET devices on flexible organic substrates and have achieved some achievements. However, there are still limitations due to the poor quality of ferroelectric gate GFET prepared on flexible organic substrate and the large operating voltage of organic ferroelectric GFET. In the early studies, for example, the organic ferroelectric gate material poly(vinylidene-fluoride-co-trifluoroethylene) (P(VDF-TrFE)) reported in the literature requires an operating voltage of at least ±4 V, but usually ±40 V [10,11]. The large operating voltage cannot meet the demand for low power consumption of the device, and with the gradual maturation of the development of flexible inorganic ferroelectric films in recent years, people gradually focus on high-performance inorganic crystal film ferroelectric gate GFET.

For flexible devices, flexible substrates and high-quality flexible dielectric films are critical. In recent years, some high-quality flexible inorganic oxide thin films have been made by directly depositing them on a mica substrate or transferring them to a flexible substrate by etching the buffer layer that grows between the thin film and the rigid substrate [12,13]. Related research shows that the obtained flexible inorganic ferroelectric films have excellent ferroelectric and piezoelectric properties and excellent development prospects in ferroelectric memory, capacitors, and piezoelectric sensors [14,15,16]. Meanwhile, high-performance inorganic ferroelectric films as gates for GFET can significantly reduce the operating voltage [9,17]. Hu et al. have produced flexible GFET devices on fluorophlogopite (F-mica) using relaxor ferroelectric film of (Pb_0.92_La_0.08_)(Zr_0.52_Ti_0.48_)O_3_ (PLZT(8/52/48)), it can tune the doping state of graphene in a wide range due to the good flexoelectric effect of PLZT(8/52/48) and can be applied to bending detection [18]. However, in the application of photodetector or non-volatile memory, instead, a more stable graphene-doped state of bending is needed. 

In this paper, we designed a flexible GFET device with the ferroelectric gate of (Pb_0.92_La_0.08_)(Zr_0.30_Ti_0.70_)O_3_ (PLZT(8/30/70)). The polarization of the PLZT(8/30/70) ferroelectric gate presents a relatively stable state under bending deformation, which is attributed to the neutralization of its weak flexoelectric effect and the piezoelectric effect. This makes the PLZT(8/30/70) gate GFET devices exhibit a stationary *V*_Dirac_, a specific graphene-doped state, in different bending states. At the same time, due to the intervention of the inorganic ferroelectric gate, the *V*_Dirac_ of the GFET device is stabilized within 2 V, achieving a smaller operating voltage. This will be very beneficial to its application in flexible devices, such as photodetectors or non-volatile memory, among others.

## 2. Materials and Methods

### 2.1. Materials and Fabrication

A buffer layer of SrTiO_3_ (STO) with a thickness of 35 nm was first epitaxially grown on the F-mica substrate using a pulsed laser deposition system at 800 °C with an oxygen pressure of 7 Pa oxygen pressure. Then the electrode layer of La_0.67_Sr_0.33_MnO_3_ (LSMO) with a thickness of 35 nm was deposited on the STO buffer using the same deposition system at 750 °C with 30 Pa oxygen pressure. Finally, the PLZT(8/30/70) ferroelectric film with 500 nm thickness was deposited on them at 650 °C and 20 Pa oxygen pressure and annealed in pure oxygen at 350 Torr for 15 min at the growth temperature. After the ferroelectric film PLZT(8/30/70) was prepared, the graphene (Muke Nano, Nanjing, China) was transferred to the PLZT(8/30/70) by wet etching, and a GFET with a channel of 50 μm was obtained by photolithography. The PLZT(8/30/70) gated GFET can achieve great flexibility when F-mica is stripped to a thickness of tens of microns. The specific graphene transfer process is as follows: A layer of polymethyl methacrylate (PMMA) was spin-coated onto a commercial graphene/copper sheet of 1 × 1 cm^2^ size and baked at 60 °C for 30 min in air. It was cut into small sheets of size 0.2 × 0.6 cm^2^ to expose the edges of the copper sheet for etching. The graphene sample was then immersed in the copper etching solution (ferric chloride solution) to remove the copper sheet. After the copper was completely dissolved, the PMMA/graphene was rinsed several times with deionized (DI) water. The graphene sample was then transferred to the PLZT(8/30/70) with prefabricated Pt source and drain electrodes and baked in air at 80 °C for half an hour to remove moisture. Finally, PMMA was removed from the graphene with acetone, and then the sample was washed with ethanol to remove residues from the graphene surface.

### 2.2. Material Characterization

The phase of the STO, LSMO, and PLZT(8/30/70) thin films was characterized using the high-resolution X-ray diffraction system. For the test of ferroelectricity, the electrode of Pt with a thickness of 80 nm and a side length of 200 mm was deposited on the PLZT(8/30/70) sample by shadow mask sputtering. The polarization curves were measured using a ferroelectric analyzer with a frequency of 1 kHz. The laser Raman spectroscopy was used to investigate the structure of graphene, and a Keithley 4200 semiconductor analyzer was used to characterize the transport properties of GFETs.

## 3. Results and Discussion

### 3.1. Polarization Characteristics of PLZT(8/30/70)

The structure of PLZT(8/30/70)/LSMO/STO multilayer film is shown in Figure 1a, and the XRD patterns of PLZT(8/30/70)/LSMO/STO films on F-mica substrate are shown in Figure 1b. Only the (111) peak can be found for both STO and LSMO thin films, indicating the STO is a good buffer layer to fabricate high-quality dielectric thin films on (001) F-mica substrate. Only the (111) peak of PLZT(8/30/70) film can be seen, indicating that PLZT(8/30/70) film with high crystallinity is obtained. To verify its ferroelectricity, the polarization-voltage (*P*-*V*) hysteresis loops with different maximum voltages of 2–10 V were measured after about 80 nm of Pt was coated on the surface of LSMO and PLZT(8/30/70) films, respectively, as shown in Figure 1c. It can be seen that the *P*-*V* hysteresis loop shows a typical ferroelectric hysteresis curve, which indicates that the PLZT(8/30/70) film has good ferroelectricity. However, the coercive voltages *V*_c_+ and *V*_c_− are not equal. This can be attributed to the difference in contact potentials or the existence of a built-in electric field at the interface of the PLZT(8/30/70) film [19]. The low *V*_c_+ of about 2 V indicates that 2 V is enough to switch the direction of polarization in PLZT(8/30/70) films and then switch the doping state of graphene at the GFET channel. It shows low leakage current density *J* of approximately 3 × 10^−7^ A·cm^−2^ at ±2 V, as shown in Figure 1d. The high resistivity of PLZT(8/30/70) film makes it a good gate material for GFET devices.

As the polarization performance of the ferroelectric gate is very important in the ferroelectric gate GFET, the polarization of the PLZT(8/30/70) ferroelectric gate under different bending deformations was investigated first. Figure 2a shows the *P*-*V* hysteresis loops of PLZT(8/30/70) under a flat state (no bend) and the curvature radii of 12 mm, 10 mm, 8 mm, and 6 mm with a 2 V sweep voltage. It can be seen that the *P*-*V* loop moves slightly downward as the upward bending curvature increases. Comparing the *P*-*V* loops under a flat state and bending state of 6 mm curvature radius, it can be seen that the overall movement of the *P*-*V* loop is very small. Figure 2b shows the values of coercive voltage *V*_c_, remanent polarization *P*_r_, and maximum polarization *P*_max_ corresponding to the curves in Figure 2a. It can be seen that both *V*_c_+ and *V*_c_− increase linearly as the upward bending curvature increase, while the values of *P*_r_+, *P*_r_−, *P*_max_+, and *P*_max_− decrease linearly. The same trend can be seen when the sweep voltage is 10 V in Figure 2c. The details of *V*_c_+, *V*_c_−, *P*_r_+, *P*_r_−, *P*_max_+, and *P*_max_− under different bending states at 2 V and 10 V can be seen in Figure 2b,d, respectively. At a 2 V sweep voltage, the change in *V*_c_+ and *V*_c_− is about 0.02 V and 0.03 V, the reduction in *P*_r_+ and *P*_r_− is 0.57 μC·cm^−2^ and 0.29 μC·cm^−2^, respectively, and the change in *P*_max_+ and *P*_max_− is 0.56 μC·cm^−2^ and 0.51 μC·cm^−2^, respectively. Compared to their value, the change rate of *V*_c_+, *V*_c_−, *P*_r_+, *P*_r_−, *P*_max_+, and *P*_max_− is about 4%, 2%, 5%, 8%, 3%, and 5%, respectively. Even at 10 V applied voltage, the change rate of *V*_c_+, *V*_c_−, *P*_r_+, *P*_r_−, *P*_max_+, and *P*_max_− is about 6%, 3%, 9%, 6%, 5%, and 8%, respectively. All of them are not more than 10%. These results indicate that the ferroelectric PLZT(8/30/70) films maintain a relatively stable state under bending deformation, although the polarization curves are slightly shifted along the polarization axis.

In general, the bending polarization of a flexible ferroelectric film can be attributed to the flexoelectric effect [20]; the polarization induced by the flexoelectric effect can be expressed as *P* = *χE* + *μ ∂u*/*∂x*, where *χ*, *E*, *μ*, and *∂u*/*∂x* are the dielectric polarization rate, applied electric field, flexoelectric coefficient, and the strain gradient during film bending. The polarization direction in the flexoelectric effect is the same as the bending direction, as shown in Figure 3a,b. Therefore, if only the flexoelectric effect is considered, the *P*-*V* loop of the film will show a downward shift when bending downward and an upward shift when bending upward. However, this is contrary to our results that the *P*-*V* curve moves down when the sample is bent upward, as shown in Figure 2. Therefore, it is not appropriate to only use the flexoelectric effect to explain the polarization in PLZT(8/30/70) under a bending state; there must be other mechanisms affecting the change of polarization of PLZT(8/30/70) ferroelectric films during bending. The easiest thing to think of is the piezoelectric effect because ferroelectric materials belong to piezoelectric materials and have a piezoelectric effect. When a material is deformed by an external force in a certain direction, its internal polarization occurs, and positive and negative charges appear on its two opposite surfaces. When the external force is removed, it will return to its uncharged state. For flexible PLZT films prepared on Mica, although Mica has been peeled to a thickness of only a few microns, the thickness of Mica is much greater than that of PLZT. Therefore, during the upward bending process, the PLZT film is subjected to tensile stress in the horizontal direction and compressive stress in the vertical direction of the sample. As the polarization direction detected by us is the vertical direction, only the conversion of the vertical direction is considered. The polarization direction generated by upward bending inside PLZT is downward, as shown in Figure 3c. In contrast, when the sample is bent downward, the PLZT film as a whole is subjected to compressive stress in the horizontal direction and tensile stress in the vertical direction. As the polarization direction detected by us is vertical, only the transformation of the vertical direction is considered. The polarization direction generated by upward bending inside PLZT is upward, as shown in Figure 3d. 

In addition, the repeatability of polarizations in different bending states also proves that these polarizations can be recovered without external forces. Combined with the piezoelectric effect, the polarization shift of PLZT(8/30/70) can be reasonably explained. It can be seen that the direction of the polarization induced by the piezoelectric effect is opposite to that induced by the flexoelectric effect. Therefore, under the action of the flexoelectric effect and piezoelectric effect, the total polarization of the PLZT(8/30/70) film under bending deformation is in a mutually canceled state, so the overall *P*-*V* loops of PLZT(8/30/70) shows a relatively stable state. In contrast, the slightly downward trend of the *P*-*V* loops indicates that the stress gradient effect is slightly dominant in PLZT(8/30/70) film.

### 3.2. Electrical Transport Properties of PLZT(8/30/70) Gated GFET

Further, based on the ferroelectric PLZT(8/30/70) film, a flexible GFET was fabricated. The structure of the flexible GFET with PLZT(8/30/70) gate can be seen in Figure 4a, where the LSMO film is used as the back gate electric, Pt is used as the source and drain electrode, and graphene acts as the channel. It can be seen from the Raman spectrum of the graphene channel in Figure 4b that the intensity of the 2D peak is greater than that of the G peak, indicating the graphene is a monolayer. 

In order to obtain accurate and reliable data, it is necessary to remove the effects of moisture and impurities at the channel interface. It has been shown that vacuum annealing can effectively remove the external hysteresis caused by contaminant molecules adsorbed on the channel of ferroelectric gate GFET devices, and this removal is permanent [9]. Therefore, all GFET samples were annealed in a vacuum below 10^−5^ mbar for about 1 day prior to testing. The source-drain current (*I*_D_) vs. gate voltage (*V*_G_) and gate leakage current (*I*_G_) vs. *V*_G_ for a representative PLZT(8/30/70) gate GFET with fixed drain bias of *V*_D_ = 1 V are shown in Figure 4c. A good *I*_D_-*V*_G_ curve (black line) can be seen with a Dirac point (*V*_Dirac_) of 1.06 V, and the positive *V*_Dirac_ indicates that the graphene is p-doped under the origin polarization state of the PLZT(8/30/70) gate [21]. Meanwhile, the low *I*_G_ (red line) of less than 10 μA was obtained due to the low *J* of PLZT(8/30/70), as discussed in Figure 1d. 

To investigate the electrical transport properties of the flexible PLZT(8/30/70) gated GFET under a bending state, the F-mica substrate was mechanically stripped to tens of microns to obtain good flexibility, and then the GFET device was transferred onto polyimide tape for bending tests. Figure 4d shows the *I*_D_-*V*_G_ curves of the flexible ferroelectric PLZT(8/30/70) gate GFET at different radii of curvature of 12 mm, 10 mm, 8 mm, and 6 mm. It can be seen that the *I*_D_-*V*_G_ curves shift towards the positive direction with increasing curvature, but the overall movement is not large, especially in the negative part. From the detailed movement of *V*_Dirac_ in Figure 4c, it can be seen that the *V*_Dirac_ exhibits a consistent increase from 1.06 V in the flat state to 1.18 V at a bending radius of 6 mm, and the shift of *V*_Dirac_ is almost linearly related to the decrease of the bending radius. However, the total shift rate of *V*_Dirac_ is about 11%, a pretty small change compared to relaxor ferroelectric PLZT(8/52/48) gate GFET in our previous work [18].

Based on the previous study of the ferroelectric gate GFET during bending, the shift of *V*_Dirac_ should be mainly due to the effect of the PLZT(8/30/70) gate on graphene at the channel during bending [18]. It is because the effect of graphene at a millimeter scale bending radius is too small to ignore, according to the report of Kundalwal et al. [22]. In order to further analyze the relationship between the doped state of graphene and PLZT(8/30/70) polarization under bending, the effect of mechanical bending on the doping of graphene is quantitatively analyzed. According to our previous report, the charge carrier change (Δ*Q*) in graphene under bending can be expressed as [18]: ∆*Q* = *C*_g_∙∆*V*_Dirac_, where *C*_g_ is the capacitance of PLZT(8/30/70) gate (about 4.3 μF·cm^−2^ as shown in Figure 4f, and the static value of dielectric constant for PLZT(8/30/70) is about 2400 measured with Agilent E4980A LCR impedance analyzer at 1 kHz and 0 V). The total change of *V*_Dirac_ for upward bending is 0.12 V, as shown in Figure 4d, which means ∆*V*_Dirac_ = 0.12 V. Therefore, when the sample is bent from ∞ to 6 mm, the Δ*Q* can be calculated as about 0.516 μC·cm^−2^, which is very close to the polarization change of Δ*P*_r_ in Figure 2b. Based on these calculations and discussions, it is clear that the movement of *V*_Dirac_ of PLZT(8/30/70) gate GFET under bending is mainly due to the polarization change of the ferroelectric PLZT(8/30/70) film under bending. On this basis, the relatively small change in *V*_Dirac_ can be explained as the polarization cancellation of PLZT(8/30/70) under the combined influence of the flexoelectric effect and piezoelectric effect under bending deformation. The relatively stable of *V*_Dirac_ for ferroelectric PLZT(8/30/70) gate GFET than relaxor ferroelectric PLZT(8/52/48) gate GFET under bending deformation indicated that ferroelectric PLZT(8/30/70) gate GFET is more suitable for the application of flexible electronic devices, such as flexible photoelectric detection combined with photoelectric materials. In addition, the bending stability of the GFET devices is highly reproducible, which indicates that the all-inorganic flexible GFET exhibits excellent flexural fatigue resistance, as shown in Figure 5a. The *I_D_*-*V_G_* curves of the GFET bending cycles largely overlap between 1 and 500 at a bending radius of 10 mm. The *V*_Dirac_ values of these curves are extracted and summarized in Figure 5b. The *V*_Dirac_ hardly changes with increasing bending cycles, indicating that the PLZT(8/30/70) gate flexible GFET has stable flexural fatigue characteristics.

## 4. Conclusions

In conclusion, we have prepared high-quality flexible PLZT ferroelectric film on an F-mica substrate with the multilayer film structure of PLZT(8/30/70)/LSMO/STO. The *P*-*V* hysteresis curve is observed to be displaced along the polarization axis as the bending curvature change, and unlike the flexoelectric effect, there is an opposite displacement direction. This is attributed to the piezoelectric effect of the film induced by bending, which affects the polarization of the film. Based on the flexible PLZT(8/30/70) film, we prepared a flexible ferroelectric gate GFET device. Corresponding to the polarization displacement of the PLZT(8/30/70) film, there is also an opposite direction of *V*_Dirac_ displacement of the GFET device compared to the *V*_Dirac_ displacement of the flexible relaxor ferroelectric PLZT(8/52/48) gate GFET caused by the flexoelectric effect. The displacement of *V*_Dirac_ is very small due to the combined effect of the piezoelectric effect and flexoelectric effect. The stability of flexible PLZT(8/30/70) gate GFET makes it promising for applications in related flexible electronic devices, such as a flexible photodetector or non-volatile memory, among others.

## Figures and Tables

**Figure 1 materials-16-03798-f001:**
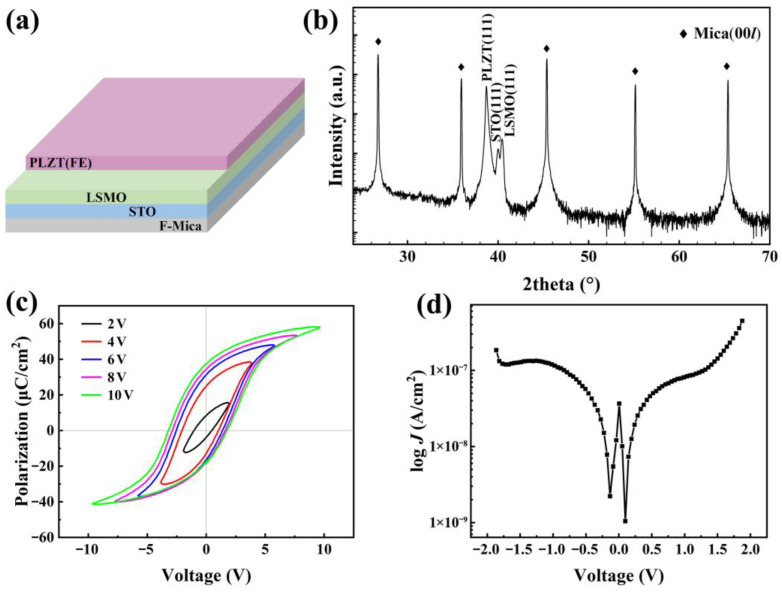
(**a**) The multilayer film structure of PLZT(8/30/70)/LSMO/STO on F-mica. (**b**) *θ*-2*θ* scans of PLZT(8/30/70)/LSMO/STO. *P*-*V* loops (**c**) and leakage current density (**d**) of the PLZT(8/30/70) on F-Mica.

**Figure 2 materials-16-03798-f002:**
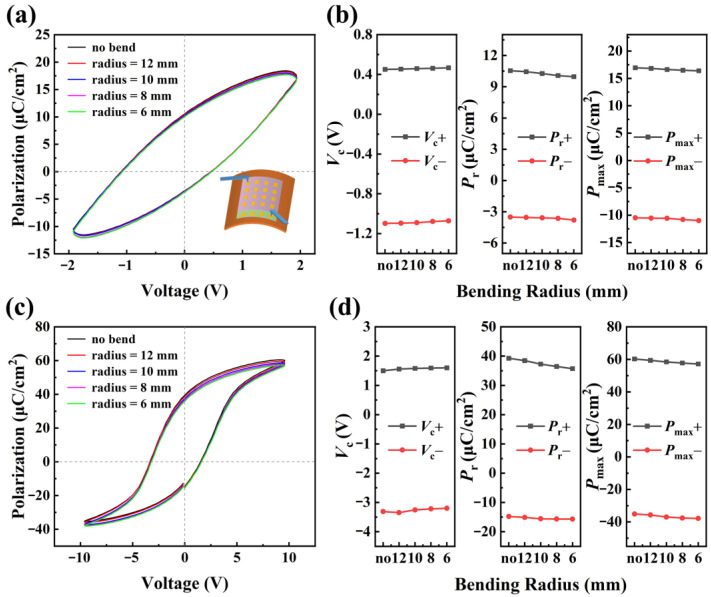
(**a**) *P*-*V* loops of PLZT(8/30/70) with 2 V sweep voltage under different upward bending radii. The inset is a schematic of the measurement. (**b**) *V*_c_, *P*_r_, and *P*_max_ of loops in (**a**). (**c**) *P*-*V* loops of PLZT(8/30/70) with 10 V sweep voltage under different upward bending radii. (**d**) *V*_c_, *P*_r_, and *P*_max_ of loops in (**c**).

**Figure 3 materials-16-03798-f003:**
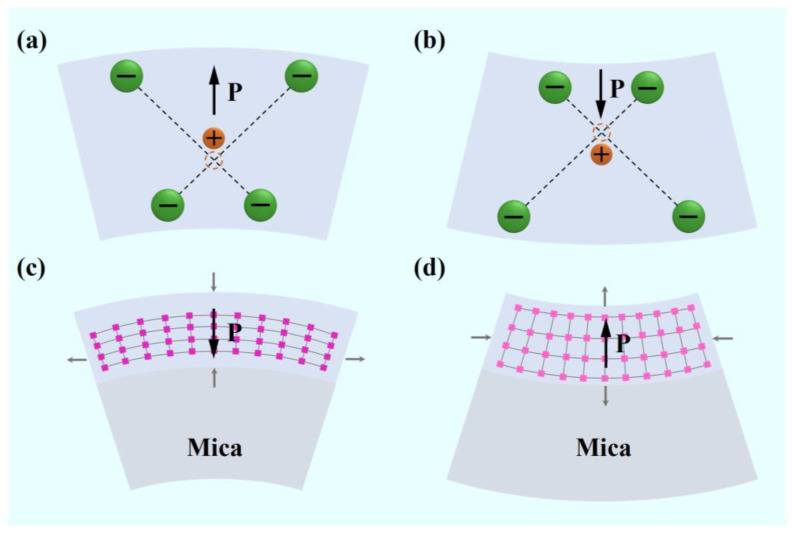
Schematic diagram of polarization induced by flexoelectric effect during upward (**a**) and downward (**b**) bending in PLZT(8/30/70). The stress and polarization in PLZT(8/30/70) during upward bending (**c**) and downward (**d**) bending.

**Figure 4 materials-16-03798-f004:**
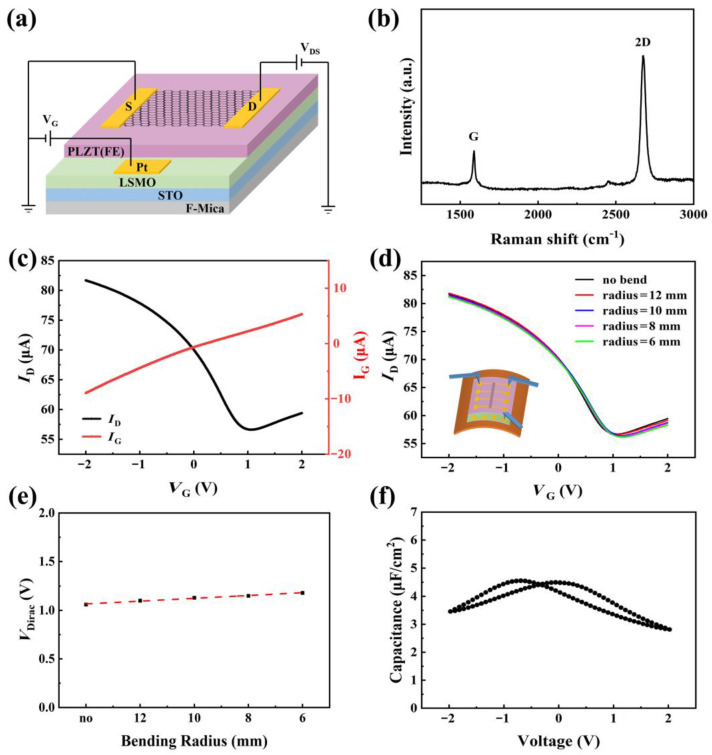
(**a**) The diagram of GFET with PLZT(8/30/70) as gate and the electrical transport test schematic. (**b**) Raman spectrum of the graphene channel on GFET. (**c**) *I*_D_-*V*_G_ (black) and *I*_G_-*V*_G_ (red) curves of PLZT(8/30/70) gate GFET. (**d**) *I*_D_-*V*_G_ curves of GFET under different upward bending states. (**e**) *V*_Dirac_ obtained from (**d**). (**f**) The capacitance of the PLZT(8/30/70) under the voltage of 2 V.

**Figure 5 materials-16-03798-f005:**
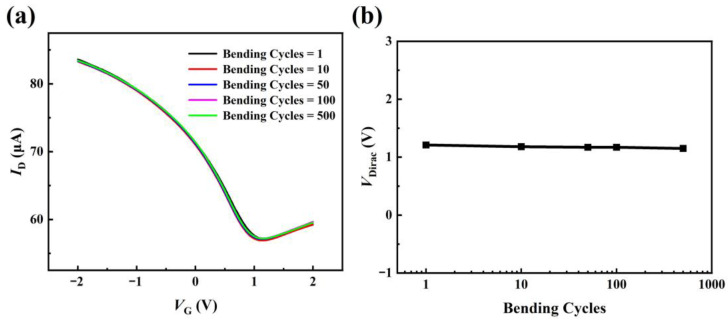
(**a**) *I_D_-V_G_* curve of GFET after multiple bending in the upward bending state. (**b**) Variation curve of *V*_Dirac_ after multiple bending.

## Data Availability

Not applicable.

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
