# Peer review of "Bending Stability of Ferroelectric Gated Graphene Field Effect Transistor for Flexible Electronics"

_materials, 2023, doi:10.3390/ma16103798_

Round 1

Reviewer 1 Report

The manuscript reports on the impact of the mechanical stress on ferroelectric properties of PLZT films and Dirac point in GFET. The work deserves publication, but several issues need to be addressed. My criticisms are listed below.

- p. 3 "However, the coercive voltages Vc+ and Vc− are not equal, this can be attribute to the exist of built-in electric field at the interface of PLZT(8/30/70) film [19]".

The most straightforward reason of different coercive voltages is the contact difference of potentials (Volta potential).

- p. 3 "The low Vc+ of about 2 V indicates that 2 V is enough to switch the direction of polarization in PLZT(8/30/70) films"

This statement is non-correct. In this work, 2 V is enough to switch the direction of polarization in a part of film. From Fig. 1c it is evident that ~ 6 V is required to switch the polarization in the most part of film.

- How the authors explain why positive and negative remanent polarizations are of different values? Is there any physical reason?

 In my opinion, the dependence of sum of absolute values of positive and negative remanent polarizations on bending radius and the same dependence for maximal polarizations would be more informative than the dependences for positive and negative polarizations separately.

- p. 4 "In general, the bending polarization of a flexible ferroelectric film can be attribute to the flexoelectric effect".

There is a more straightforward explanation of the observed effect - the impact of stress that causes changes in lattice constants, whereas the flexoelectric effect originates from the stress gradient, and its impact is weaker.

- p. 5-6 "In order to obtain accurate and reliable data, it is necessary to remove moisture and impurities at the channel interface."

Measurements were carried out in vacuum? Otherwise, it should be taken into account that under room conditions, the adsorbate on graphene is appeared within 10 minutes after its removal.

- Why do the authors not demonstrate the effect of polarization reversal on the transistor Id-Vg curve? Is it possible to show the hysteresis of the Id-Vg curve, which should be observed during polarization reversal?

Author Response

Reviewer #1 Comments:

The manuscript reports on the impact of the mechanical stress on ferroelectric properties of PLZT films and Dirac point in GFET. The work deserves publication, but several issues need to be addressed. My criticisms are listed below.

  1. p. 3 "However, the coercive voltages Vc+ and Vc are not equal, this can be attribute to the exist of built-in electric field at the interface of PLZT(8/30/70) film [19]".

The most straightforward reason of different coercive voltages is the contact difference of potentials (Volta potential).

Response: Thanks for your suggestion. You are right, the contact of potentials is the most straightforward reason of different coercive voltages. But the model with a non-ferroelectric layer (non-switching layer as shown in Fig.13 in Ref.19) is also a reasonable explanation for this phenomenon. So, I revised this sentence as "However, the coercive voltages Vc+ and Vc are not equal, this can be attribute to the difference of contact potentials or the exist of built-in electric field at the interface of PLZT(8/30/70) film [19]".

  1. p. 3 "The low Vc+ of about 2 V indicates that 2 V is enough to switch the direction of polarization in PLZT(8/30/70) films"

This statement is non-correct. In this work, 2 V is enough to switch the direction of polarization in a part of film. From Fig. 1c it is evident that ~ 6 V is required to switch the polarization in the most part of film.

Response: Thanks for your suggestion. Form the Fig.1c in manuscript, it can be seen that all the Vc+ is less than 2.5V. Here, I mean the voltage of 2V can switch state of negative polarization to positive polarization, but not all the dipoles are switched at this voltage. The ~6V voltage you indicate may be the voltage that all the dipoles are switched.

  1. How the authors explain why positive and negative remanent polarizations are of different values? Is there any physical reason?

In my opinion, the dependence of sum of absolute values of positive and negative remanent polarizations on bending radius and the same dependence for maximal polarizations would be more informative than the dependences for positive and negative polarizations separately.

Response: Thanks for your suggestion, but I don’t agree with you. This can be explained using a similar model to ref.19. The polarization induced by bending can be regarded as the non-switching layer or hard-switching layer, which is hard to polarize by voltage. The potential difference in this polarized layer can be regarded as a built-in electric field, and it does not affect the sum of absolute values of positive and negative remanent polarizations, but causes the curve to move as a whole.

  1. p. 4 "In general, the bending polarization of a flexible ferroelectric film can be attribute to the flexoelectric effect".

There is a more straightforward explanation of the observed effect - the impact of stress that causes changes in lattice constants, whereas the flexoelectric effect originates from the stress gradient, and its impact is weaker.

Response: Thanks for your suggestion, but I don’t agree with you. The flexoelectric effect originates from the strain gradient not stress gradient, and the most obvious phenomenon when the sample is bent is the linear strain of the material. Normally, the magnitude of the flexoelectric effect is very small in a solid, but it is very important for thin films, especially at the nanoscale, its contribution to the overall thermodynamics of a solid may become significant or even dominant at the nanometer scale.

  1. p. 5-6 "In order to obtain accurate and reliable data, it is necessary to remove moisture and impurities at the channel interface."

Measurements were carried out in vacuum? Otherwise, it should be taken into account that under room conditions, the adsorbate on graphene is appeared within 10 minutes after its removal.

Response: This is a good question and thanks for your suggestion. Although our measurements are not conducted in a vacuum, we have taken effective measures to address this issue. As you said, the contaminant molecules adsorbed in graphene cause external hysteresis in ferroelectric gate GFET devices, while the oxygen-deficient layer surface of the ferroelectric gate causes internal hysteresis, and the coexistence of these two hysteresis effects causes a drift in the Dirac point. However, it has been shown that vacuum annealing can effectively solve this problem. As the annealing time increases, the ferroelectric film transitions from the coexistence of intrinsic and extrinsic hysteresis states to an anti-hysteresis state induced by the oxygen-deficient surface layer. Vacuum annealing can effectively eliminate the external hysteresis, and this elimination is permanent. No external hysteresis is observed even when the ferroelectric gate GFET device is measured again in air after vacuum annealing. Therefore, we annealed all GFET samples in a high vacuum environment below 10-5 mbar for about 24 hours prior to testing to eliminate possible external hysteresis from contaminant molecules adsorbed by graphene on the GFET devices.

We have revised and added the corresponding explanations in the revised manuscript.

In the revised manuscript, pages 4-5. “In order to obtain accurate and reliable data, it is necessary to remove the effects of moisture and impurities at the channel interface. It has been shown that prolonged vacuum annealing can effectively remove the external hysteresis caused by contaminant molecules adsorbed by graphene on ferroelectric gate GFET devices, and that this removal is permanent.[9]”

  1. Why do the authors not demonstrate the effect of polarization reversal on the transistor Id-Vg curve? Is it possible to show the hysteresis of the Id-Vg curve, which should be observed during polarization reversal?

Response: Thanks for your suggestion. The single sweep model has included the reversal of negative polarization to positive conversion of ferroelectric gate, corresponding to the left and right side of VDirac with Id-Vg curves. Do you mean the double sweep? The focus of this manuscript is on the study of the transport properties of PLZT(8/30/70) ferroelectric graphene field effectors by bending deformation, and we are mainly concerned with VDirac changes, so we are not doing a double sweep.

Reviewer 2 Report

This work deals with the Bending Stability of the Ferroelectric Gated Graphene Field Effect Transistors for Flexible Electronics. Its review is very interesting work in the related field for potential readers. However, there is not clear something for publication and I recommend the author should improve the manuscript.

 1. In this work, you selected PLZT and graphene. In another group, they also have been studied for electrical applications. What is motivation? I recommend you would add your originality or motivation in the introduction.

 2. After bending, how about the durability of the sample? If possible, after bending several times, could you show an SEM image of the samples?

 3. Compared to other groups’ data, where your sample’s performance is level?

Author Response

Reviewer #2 Comments:

This work deals with the Bending Stability of the Ferroelectric Gated Graphene Field Effect Transistors for Flexible Electronics. Its review is very interesting work in the related field for potential readers. However, there is not clear something for publication and I recommend the author should improve the manuscript.

Response: Thank you very much for taking the time to read and provide valuable suggestions and questions.

  1. In this work, you selected PLZT and graphene. In another group, they also have been studied for electrical applications. What is motivation? I recommend you would add your originality or motivation in the introduction.

Response: This is a good question and thanks for your suggestion. Conventional Si-based GFET devices have the disadvantage of large operating voltage and cannot be flexible. Organic ferroelectric gate GFET devices can be flexible but still have a large operating voltage. The inorganic ferroelectric gate GFET devices can solve both problems at the same time. Our group has previously prepared relaxed ferroelectric PLZT (8/52/48) gate GFET devices on STO substrates to obtain operating voltages below 2 V, while detecting interfacial dipole interactions with the same. Also relaxed ferroelectric PLZT (8/52/48) gate GFET flexible devices were prepared by on F-Mica substrates, but due to the good flexible electrical effect of PLZT (8/52/48), it can modulate the doping state of graphene over a wide range and therefore does not obtain a more stable Dirac point. However, in photodetector or non-volatile memory applications, a more stable graphene-doped bending state is instead required. Therefore, we designed a flexible GFET device with ferroelectric PLZT (8/30/70) as the gate, and it was shown that the device exhibits a fixed VDirac, a special graphene-doped state, in different bending states. This will be very beneficial for its application in flexible devices.

We have revised and added the corresponding explanations in the revised manuscript.

In the revised manuscript, pages 1-2. “Therefore, in order to solve the flexibility problem, researchers have tried to prepare ferroelectric gate GFET devices on flexible organic substrates and have achieved some achievements. … The large operating voltage cannot meet the demand for low power consumption of the device, and with the gradual maturation of the development of flexible in-organic ferroelectric films in recent years, people gradually focus on high-performance inorganic crystal film ferroelectric gate GFET….”

In the revised manuscript, pages 2. “At the same time, due to the intervention of the inorganic ferroelectric gate, the VDirac of the GFET device is stabilized within 2V, achieving a smaller operating voltage.”

  1. After bending, how about the durability of the sample? If possible, after bending several times, could you show an SEM image of the samples?

Response: This is a good question. We also performed a bending fatigue test to ensure the applicability of the device. We chose a bending radius of 10 mm and bent the sample 10, 50, 100 and 500 times to test its I-V characteristics, as shown in Fig. 5a. It can be seen that the sample remains very stable after repeated bending. And as shown in Fig. 5b, the Dirac point of the sample is almost not shifted. Therefore, it can be shown that our PLZT ferroelectric gate GFET device has good durability.

We have revised and added the corresponding explanations in the revised manuscript.

In the revised manuscript, pages 7. “In addition, the bending stability of the GFET devices is highly reproducible, which indicates that the all-inorganic flexible GFET exhibits excellent flexural fatigue resistance, as shown in Fig. 5a. The ID-VG curves of the GFET bending cycles largely overlap between 1 and 500 at a bending radius of 10 mm. The VDirac values of these curves are extracted and summarized in Fig. 5b. The VDirac hardly changes with in-creasing bending cycles, indicating that the PLZT(8/30/70) gate flexible GFET has a stable flexural fatigue characteristics.”

  1. Compared to other groups’ data, where your sample’s performance is level?

Response: This is a good question. Most of the current ferroelectric gate GFET devices are not flexible, and there are already flexible organic ferroelectric gate devices with large operating voltages (~20V). Only the GFET flexible device with relaxed ferroelectric PLZT (8/52/48) as the gate previously prepared by our group, the Dirac point is controlled at 0.72V, but VDirac has a large offset (~0.32V) in the bending state, and the offset is about 45% of itself. In contrast, the GFET flexible device designed in this paper with ferroelectric PLZT (8/30/70) as the gate controls the Dirac point at 1.08V, and the offset in the bending state is 0.12V, about 11%, which greatly improves the stability of the device.

Reviewer 3 Report

The paper deals with the fabrication of a FET using ferroelectric gate dielectric. The paper shows important concerns.

1.- Based on Fig. 1a the authors conclude that PLZT is obtained with high crystallinity. However, the peaks related with the substrate are too high which screen the full pattern. It is recommended to measure the film avoiding the substrate interference.

2.- Based on Fig. 1d the authors conclude that the PLZT is a good material for gate applications. Actually, the film shows a high leakage. According with section 2.1 the PLZT film thickness is 500 nm. For such thickness a current density of about 10^-7 A/cm^2 is not small. Also, as can be seen from Fig. 4c the gate current and the drain current are similar.
Also, as observed from Fig. 1d the gate PLZT current density starts to increase around ±2V, this value implies an electric field of about 4E4 V/cm which too small for gate dielectric applications. Finally, could the authors explain the presence of the two minimum peaks around zero bias?

3.- Regarding the transistor behavior, the authors only show the transfer characteristic. It is necessary to show the output characteristic in order to properly analyze the device behavior.
Also the authors indicate that the area normalized gate capacitance is about 4.3 mF/cm^2 (Fig. 4f). This value seems unrealistic… which is the dielectric constant of the PLZT film?

Author Response

Reviewer #3 Comments:

The paper deals with the fabrication of a FET using ferroelectric gate dielectric. The paper shows important concerns.

Response: Thank you very much for taking the time to read and provide valuable suggestions and questions.

  1. Based on Fig. 1a the authors conclude that PLZT is obtained with high crystallinity. However, the peaks related with the substrate are too high which screen the full pattern. It is recommended to measure the film avoiding the substrate interference.

Response: Thanks for your suggestion. It is diffcult to do the XRD scanning avoiding the substrate interference, as the PLZT(8/30/70) was grown on the LSMO/STO/Mica substrate and cannot peel. But it can be seen that there are no other PLZT peaks in the whole sweep area except the (111) peak of PLZT(8/30/70), which is enough to indicate that the quality of the grown PLZT(8/30/70) film is very good. In fact, when the PLZT is not high crystallinity, there will be some miscellaneous peaks near 30 and 50 degrees.

  1. Based on Fig. 1d the authors conclude that the PLZT is a good material for gate applications. Actually, the film shows a high leakage. According with section 2.1 the PLZT film thickness is 500 nm. For such thickness a current density of about 10^-7 A/cm^2 is not small. Also, as can be seen from Fig. 4c the gate current and the drain current are similar.

Also, as observed from Fig. 1d the gate PLZT current density starts to increase around ±2V, this value implies an electric field of about 4E4 V/cm which too small for gate dielectric applications. Finally, could the authors explain the presence of the two minimum peaks around zero bias?

Response: Thanks for your good questions. We say the PLZT is a good material for gate applications means the PLZT has a high polarization at a small gate voltage, and can induce a deep doping of graphene. This is very benefit to us to obtain an “V” curve of GFET. The leakage is indeed not small compared to our previous work, which may be caused by the damage of PLZT during the preparation of GFET. Thanks to the large Id ratio, this does not affect the study in this manuscript.

“The two minimum peaks around zero bias” you mentioned means the Id at VDirac? This is a good question. In fact, I have seen similar cases in other literature, but none of them have been explained. Here I think it is possible that the resistance of the graphene itself changes during the bending process.

  1. Regarding the transistor behavior, the authors only show the transfer characteristic. It is necessary to show the output characteristic in order to properly analyze the device behavior.

Also the authors indicate that the area normalized gate capacitance is about 4.3 mF/cm^2 (Fig. 4f). This value seems unrealistic… which is the dielectric constant of the PLZT film?

Response: Thanks for your suggestion. I'm not quite sure what you mean by output signal, but the main work of this manuscript is to study the stability properties of PLZT gate GFET under bending deformation for more applications of PLZT gate GFET in flexible devices.

The capacitance of PLZT was measured by ferroelectric analyzer TF 2000. The thickness and electrode are same as the PLZT gate in GFET. While, the dielectric constant of PLZT can be measure by Agilent E4980A LCR impedance analyzer. They have different value.

Reviewer 4 Report

The manuscript “Bending Stability of Ferroelectric Gated Graphene Field Effect Transistor for Flexible Electronics” claims that it was found a new design and material [Pb (0.92) La (0.08 )Zr (0.30) Ti (0.70 ) O (3)] that make a ferroelectric gated graphene field effect transistor stable against bending.

The experimental facts are convincing, but the authors’ explanations are not appropriate for the case they studied. Flexoelectric effect is a second order effect, that can manifest visibly in materials with no piezoelectric properties. Ferroelectrics are piezoelectric materials, hence the piezoelectric polarization are the main effect on device bending.

In addition, even though mica has a very small Young modulus (smaller by a factor of 20-30 with respect to the piezo- and dielectric layers), its thickness is more that 100 times larger the the thickness of the piezo- and dielectric layers. Thus the neutral layer (with 0 strain, the region where the strain reverses its sign) would lay into mica layer and the sign of strain does not change in the piezoelectric material. We also have to consider the electronic structure change in graphene due to bending. I think that this electronic structure change and piezoelectric effect manifested in the ferroelectric lead to apparent insensibility of electrical parameters that govern the charge transport in the ferroelectric gated graphene field effect transistor.

The authors need to come up with a better and a convincing explanation of their fabricated device.

Moreover, they need to work on the entire form of the manuscript – abstract, introduction, etc. - because the manuscript has many language errors.

In conclusion, at this stage I cannot recommend the manuscript for publication.

Author Response

Reviewer #4 Comments:

The manuscript “Bending Stability of Ferroelectric Gated Graphene Field Effect Transistor for Flexible Electronics” claims that it was found a new design and material [Pb (0.92) La (0.08 )Zr (0.30) Ti (0.70 ) O (3)] that make a ferroelectric gated graphene field effect transistor stable against bending.

The experimental facts are convincing, but the authors’ explanations are not appropriate for the case they studied. Flexoelectric effect is a second order effect, that can manifest visibly in materials with no piezoelectric properties. Ferroelectrics are piezoelectric materials, hence the piezoelectric polarization are the main effect on device bending.

In addition, even though mica has a very small Young modulus (smaller by a factor of 20-30 with respect to the piezo- and dielectric layers), its thickness is more that 100 times larger the the thickness of the piezo- and dielectric layers. Thus the neutral layer (with 0 strain, the region where the strain reverses its sign) would lay into mica layer and the sign of strain does not change in the piezoelectric material. We also have to consider the electronic structure change in graphene due to bending. I think that this electronic structure change and piezoelectric effect manifested in the ferroelectric lead to apparent insensibility of electrical parameters that govern the charge transport in the ferroelectric gated graphene field effect transistor.

The authors need to come up with a better and a convincing explanation of their fabricated device.

Moreover, they need to work on the entire form of the manuscript – abstract, introduction, etc. - because the manuscript has many language errors.

In conclusion, at this stage I cannot recommend the manuscript for publication.

Response: Thanks for your suggestions. You are right. On the one hand, the piezoelectric effect does exist in ferroelectric materials, and the piezoelectric should not be ignored. In fact, although the flexoelectric effect is not obvious for ferroelectrics in general, it is very obvious for thin films, especially nanoscale films. The flexoelectric effect is also very important. On the other hand, as you say, since the thickness of mica is much greater than that of the ferroelectric film, it is correct that the neutral layer (with 0 strain) of the ferroelectric film would lay into the mica layer when the sample is bent. I quite agree with you. Based on this view, when the sample is bent upward, the ferroelectric film will be subjected to compressive stress in the vertical direction as well as the horizontal direction of tensile stress. Since the polarization detected by us is mainly from the vertical direction, it can be simplified to the compressive stress of only the vertical component. In this case, the direction of polarization in the ferroelectric film is downward. Conversely, when the sample is bent downward, the ferroelectric film receives compressive stresses in the horizontal direction as well as tensile stresses in the vertical direction. When only vertical polarization is considered, it can be simplified to the tensile stress of only vertical component. In this case, the direction of polarization in the ferroelectric film is upward.

Although the result has not changed, I think it is more reasonable to use piezoelectric effect to explain according to your suggestion, so we have modified the corresponding text and picture (Fig.3) in the paper.

In addition, according to your suggestions, we have checked and corrected the grammar and spelling of the whole paper.

Round 2

Reviewer 2 Report

The authors responded to each question and comment.  Finally, I recommend font and size of the figures would be clear and large for potential readers to understand and investigate the data. 

Author Response

Reviewer #2 Comments:

The authors responded to each question and comment. Finally, I recommend font and size of the figures would be clear and large for potential readers to understand and investigate the data.

Response: Thanks for your suggestion. We have changed the font of the figures in the revised manuscript and increased the font size for potential readers to understand and investigate the data.

Reviewer 3 Report

The authors did not properly clarify the main reviewer questions.

Author Response

Reviewer #3 Comments:

The authors did not properly clarify the main reviewer questions.

Response: Thanks for your comments. For the first question of reviewer, to measure the film avoiding the substrate interference is too difficult, just as my response, the film of PLZT(8/30/70) was deposited on the substrate and forms a tight bonding force. Firstly, it is very difficult to peel the PLZT film alone, and secondly, even if it is peeled, peeling will cause the film to lose the stress effect of the bottom of the film, often changing the structure and properties of the film, which has been confirmed, and finally during the XRD test, the penetration depth of X-rays is generally in a few microns to tens of microns, which is far greater than the thickness of the film, therefore, in the field of film preparation, the method in the manuscript is usually used for the film structure testing.

For the second question of reviewer, we must admit that the leakage current of PLZT(8/30/70) film at 2 V is indeed not very low, which has a lot to do with the thickness of the film, and we have indeed encountered this situation when we studied PLZT(8/52/48) gate GFET before, but increasing the thickness can significantly improve the leakage current, we guess this may be related to its columnar growth structure. In addition, the reason why we chose 2 V as the test bias and gate voltage is, on the one hand, because the coercive electric field of PLZT(8/30/70) corresponds to a voltage of 2 V during the forward voltage sweep, and the polarization can be flipped at this voltage, and on the other hand, because the VDirac offset caused by graphene doping caused by the inherent interface charge of PLZT can be completely observed within 2 V. And with the development of low power consumption of devices, small voltage will be more advantageous for FET devices in the future.

For the second question of reviewer, this manuscript aims to explore the application possibilities of ferroelectric grid GFET in flexible devices, so it is concerned with the stability study of performance under partial base flexibility. In addition, to ensure the reliability of the area normalized gate capacitance in our manuscript, we have added static value of dielectric constant for PLZT as measured with Agilent E4980A LCR impedance analyzer at 1 kHz and 0 V, and the discussion of “……where Cg is the capacitance of PLZT(8/30/70) gate (about 4.3 μF·cm−2 as shown in Fig. 4f, and the static value of dielectric constant for PLZT(8/30/70) is about 2400 measured with Agilent E4980A LCR impedance analyzer at 1 kHz and 0 V)” have been added in manuscript.

Reviewer 4 Report

In this form I recommend the paper for publication.

Author Response

Reviewer #4 Comments:

In this form I recommend the paper for publication.

Response: Thanks for your suggestions and comments.